# Suitability Calculation for Red Spicy Pepper Cultivation (*Capsicum annum* L.) Using Hybrid GIS-Based Multicriteria Analysis

**Mladen Jurišić [1], Ivan Plaščak [1], Oleg Antonić [2] and Dorijan Radočaj [1,\*]**

[1] Department of Agricultural Engineering and Renewable Energy Sources, Faculty of Agrobiotechnical Sciences Osijek, Josip Juraj Strossmayer University of Osijek, Vladimira Preloga 1, 31000 Osijek, Croatia; mjurisic@fazos.hr (M.J.); iplascak@fazos.hr (I.P.)

[2] Department of Biology, Josip Juraj Strossmayer University of Osijek, Ulica cara Hadrijana 8/A, 31000 Osijek, Croatia; oleg.antonic@biologija.unios.hr

\* Correspondence: dradocaj@fazos.hr; Tel.: +385-31-554-879

**Abstract:** Red spicy pepper is traditionally considered as the fundamental ingredient for multiple authentic products of Eastern Croatia. The objectives of this study were to: (1) evaluate the optimal interpolation method necessary for modeling of criteria layers; (2) calculate the sustainability and vulnerability of red spicy pepper cultivation using hybrid Geographic Information System (GIS)-based multicriteria analysis with the analytical hierarchy process (AHP) method; (3) determine the suitability classes for red spicy pepper cultivation using K-means unsupervised classification. The inverse distance weighted interpolation method was selected as optimal as it produced higher accuracies than ordinary kriging and natural neighbour. Sustainability and vulnerability represented the positive and negative influences on red spicy pepper production. These values served as the input in the K-means unsupervised classification of four classes. Classes were ranked by the average of mean class sustainability and vulnerability values. Top two ranked classes, highest suitability and moderate-high suitability, produced suitability values of 3.618 and 3.477 out of a possible 4.000, respectively. These classes were considered as the most suitable for red spicy pepper cultivation, covering an area of 2167.5 ha (6.9% of the total study area). A suitability map for red spicy pepper cultivation was created as a basis for the establishment of red spicy pepper plantations.

**Keywords:** GIS; multicriteria analysis; AHP; red spicy pepper; kriging; inverse distance weighted; unsupervised classification; K-means

## 1. Introduction

Land use planning and ecological land evaluation are considered the most important tools and factors of sustainable agricultural production [1]. Investments in agriculture imply knowledge of the production potential of a particular site, with regards to crop types and optimal agroecological conditions [2]. Multicriteria analysis provides information on the optimal utilization of available resources, allowing sustainable crop management [3]. In addition to crop requirements, it is necessary to have spatial information of the farmland soil, climate and land cover properties [4,5]. The upgrade of such analysis with a spatial component of GIS allows effective planning and its realization in the field [6,7]. The GIS-based multicriteria analysis was applied successfully for crop management down to the county or municipality area [8,9]. Among the multicriteria analysis methods, AHP has become increasingly popular since it allows the integration of a large quantity of heterogeneous data, making the process of criteria weight determination straightforward, even for a large number of criteria [10].

AHP was widely used for suitability analyses in agriculture, including land-use suitability [11–13], irrigation suitability [14,15], resource evaluation [16] and organic agriculture efficiency [17].

Sustainability and vulnerability are considered as two components of land suitability. Sustainability represents a positive component and includes criteria that are necessary for optimal crop growth and development [18]. Vulnerability represents a negative component, indicating risks and threats to potential crop plantations [19]. Aside from the selection of sustainability and vulnerability criteria, also named as factors, representing the level of satisfying the selected criteria, Boolean constraint layers are commonly modelled in similar studies [9]. Constraints commonly had a support role, only determining suitable or non-suitable areas for the selected purpose due to their lack of ability to fully incorporate expert's knowledge in suitability result. In the case of non-suitability, these areas were masked from the final suitability map or classified in separate permanently non-suitable class. Application of unsupervised classification algorithms expands the capabilities of multicriteria analysis while reducing the possibilities of man-made errors in suitability calculations [20]. The most common cause of human errors in supervised classification algorithms occurs during the creation of the classification training dataset due to the user's subjectivity in the process. The process of training data creation does not apply to the unsupervised classification since its working principle is the automatic identification and creation of similar data classes, therefore allowing the retention of the classification objectivity [21]. Such an approach satisfies a demand for the establishment of management zones in precision agriculture [22]. It also enables the creation of a custom amount of suitability and non-suitability classes, according to the widely used classification specifications, most notably by the Food and Agriculture Organization (FAO) [23]. These zones are further applied as elementary units for variable rate technology in agrotechnical operations [24,25].

Red spicy pepper is an herbaceous, shrubby plant of the *Capsicum annum* L. species from the Solanaceae family. Its pharmacological and nutritional significance has been thoroughly described in [26–28]. Minced dried red spicy pepper traditionally has importance in the human diet in Croatia, being one of the most commonly used spices and the ingredient for multiple authentic products of Eastern Croatia. In 2017, red spicy pepper was cultivated on 14.2% of the area in the category of vegetables for market production in Croatia [29]. A total of 15,547 tons of red spicy pepper was produced in that category during 2017, with the average yield of 15.2 t/ha.

This study incorporates the mutually complementary methods of GIS-based multicriteria analysis, AHP, interpolation and unsupervised classification into a suitability model for red spicy pepper cultivation. Three major objectives of the study were to: (1) evaluate the optimal interpolation method necessary for modelling of criteria layers; (2) calculate the sustainability and vulnerability of red spicy pepper cultivation using hybrid Geographic Information System (GIS)-based multicriteria analysis with the analytical hierarchy process (AHP) method; (3) determine the suitability classes for red spicy pepper cultivation using K-means unsupervised classification. The purpose of the calculated results is to enable detection of the most suitable micro-locations for red spicy pepper cultivation, allowing the increase of high-profitable red spicy pepper quality and branding of related products.

## 2. Materials and Methods

### 2.1. Study Area

The municipality of Bilje is located in the lowland part of the Baranya region, in the Northeastern Croatia (Figure 1). Its most significant natural resources are agricultural land, forests and water resources. The dominant agricultural activity is farming. The climate in the study area is moist subhumid, based on the Thornweit scale and Cfwbx based on the Koppen scale. The area is generally characterized by a moderately warm, humid rainy continental climate, with a regular change in the seasons. Climate conditions in the study area have little variation and are considered homogeneous, primarily due to a relatively small municipality area (315 km$^2$).

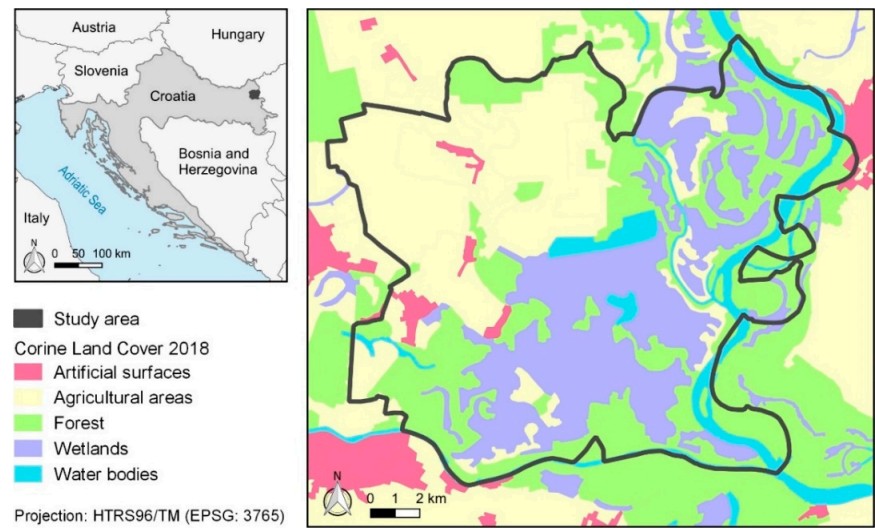

**Figure 1.** Study area with Corine Land Cover 2018 data.

## 2.2. Hybrid GIS-Based Multicriteria Analysis

The proposed methodology for red spicy pepper suitability calculation contains four major steps: pre-processing, sustainability calculation, vulnerability calculation and suitability calculation (Figure 2). Implementation of the methodology was conducted using open-source software SAGA GIS v7.3.0 (Hamburg, Germany) for data processing, interpolation and calculation, alongside QGIS v3.8.3 (Grüt, Switzerland) for data visualization. The proposed methodology integrates raster-based multicriteria analysis, so all input vector layers were rasterized. Coordinate reference system (CRS) for analysis was the Croatian Terrestrial Reference System (HTRS96/TM, EPSG: 3765). Layers which initially had different CRS were reprojected to HTRS96/TM and resampled to 50 m spatial resolution.

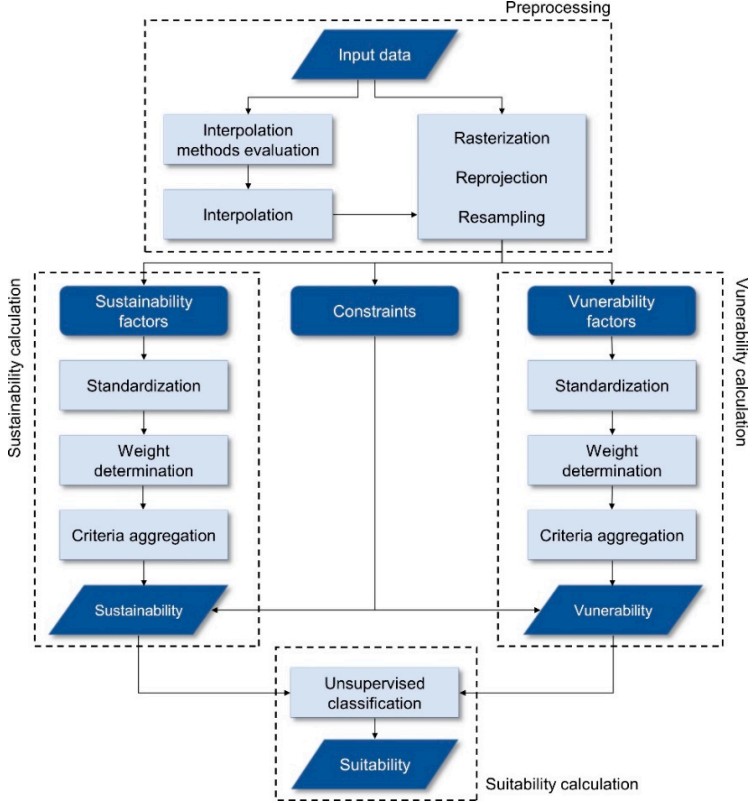

**Figure 2.** Proposed methodology workflow.

2.2.1. Input Data

Input criteria layers for red spicy pepper cultivation were collected and classified into sustainability factors, vulnerability factors and constraints. Sustainability and vulnerability factors continuously indicated an impact on sustainability and vulnerability, respectively [30]. Constraints contained Boolean data which indicated a strict possibility or inability of red spicy pepper cultivation [31]. Selected criteria for red spicy pepper cultivation are shown in Table 1. A total of 17 criteria were selected, with seven sustainability factors, five vulnerability factors and five constraints.

Sustainability factors primarily indicate soil potential for sustainable red spicy pepper cultivation. Fertility represented overall production potential based on soil type, derived from a modified Basic Soil Map of Croatia [32]. Proper soil drainage prevents precipitation stagnation near stalks, which leads to deterioration of the roots [33]. Soil organic content influences soil structure and diversity of soil organisms [34]. Two of the most important soil nutrients for red spicy pepper are potassium oxide ($K_2O$) and phosphorus pentoxide ($P_2O_5$). A sufficient amount of $P_2O_5$ enables the development of the roots and flowering. Plant growth is conditioned by the $K_2O$ amount, as well as the formation of the optimal size of fruits. Neutral and mildly acidic soils allow optimal absorption of soil nutrients for red spicy pepper. Terrain slope was selected as the last sustainability factor.

Vulnerability factors consolidated diverse factors with potentially unfavourable effects on red spicy pepper production. The unused construction area is considered vulnerable due to the potential construction of residential and industrial structures. Forests under private ownership are classified into two classes, first having high wood potential and second with dominant natural vegetation. Forests with high wood potential are considered more vulnerable than forests with dominant natural vegetation since they represent a raw material for carpentry. Lower categories of water wells protection denote only organic farming and restricted irrigation. Agricultural plots on archaeological localities can potentially be repurposed to amenities in the interest of cultural tourism.

Constraints represent areas unavailable for red spicy pepper cultivation (built-up area, water bodies and wetland area) and areas prohibited by the government (forests and first category water well protection area). The most significant constrained area is Kopački rit, a wetland area designated as a nature park. The selected criteria are dominantly quantitative, except for the qualitative fertility and drainage criteria.

**Table 1.** Criteria selected for calculation of red spicy pepper suitability.

| Criteria Type | Criteria Name | Description |
|---|---|---|
| Factors (sustainability) | Fertility | Soil type |
| | Drainage | Soil drainage level |
| | Humus | Organic matter soil content [mg/100 g] |
| | $K_2O$ | Potassium oxide soil content [mg/100 g] |
| | $P_2O_5$ | Phosphorous pentoxide soil content [mg/100 g] |
| | pH | Soil pH value |
| | Slope | Terrain slope [%] |
| Factors (vulnerability) | A | Private forests with high wood potential |
| | B | Second and third category water well protection |
| | C | Unused construction area |
| | D | Archaeological localities |
| | E | Private forests with dominant natural vegetation |
| Constraints | Built-up | Residential and infrastructural objects |
| | Forests | Forests under government jurisdiction |
| | Water | Water bodies |
| | Wetlands | Swamp area |
| | Wells | First category water well protection |

### 2.2.2. Interpolation Methods Evaluation

The interpolation of soil features was conducted as a mandatory operation in the pre-processing of the four criteria layers that are sampled in the field: humus, $K_2O$, $P_2O_5$ and pH. Due to the multiple possible options for interpolation regarding the selection of the interpolation method and its parameters, an evaluation of the interpolation methods was conducted. The selection of the optimal interpolation method was proven to have an impact on the suitability result, as the performance of interpolation methods depends on the characteristics of the input samples [35,36]. Tested interpolation methods were ordinary kriging (OK), inverse distance weighted (IDW) and natural neighbour (NN). Selected methods are mutually complementary considering the input samples' properties. OK belongs to geostatistical methods and performs best when samples possess normality, stationarity and regular sample distribution [37,38]. IDW and NN are deterministic methods that are considered more practical when samples do not possess a normal distribution and are irregularly spaced [39,40].

The interpolation accuracy test was calculated with the random split-sample method in five independent repetitions. For each repetition, the input sample consisting of 171 samples was split randomly into training (70%) and test data (30%), as used in [41]. Using the described method, we divided the input samples to 120 training and 51 test samples, which remained consistent for all five iterations. Coefficients of determination were selected as accuracy assessment values. Inner accuracy ($R^2_{inner}$) shows the relation of training data and their corresponding interpolated values. Inner accuracy represents how much an interpolation method preserves field sample values in the interpolation result, as those values are regarded as absolutely accurate. Outer accuracy ($R^2_{outer}$) shows the relation of test data and their corresponding interpolated values. Outer accuracy measures the accuracy of prediction for tested interpolation methods. Both $R^2_{inner}$ and $R^2_{outer}$ were calculated for each of five datasets and four soil features, resulting in a total of 20 tested interpolation variations. Descriptive statistics consisting of mean, coefficient of variation (CV), skewness (SK) and kurtosis (KT) were calculated for the determination of data normality and stationarity. These values served as a basis for the selection of interpolation parameters. This particularly refers to ordinary kriging, where normality and stationarity are prerequisites for its calculation. The interpolation of soil features was conducted using the most accurate interpolation method.

### 2.2.3. Sustainability and Vulnerability Calculation Method

Sustainability and vulnerability factors were separately used for the determination of the criteria impact for sustainability or vulnerability [42,43]. The identical procedure of multicriteria analysis was applied for the calculation of sustainability and vulnerability. The same constraints were used for both calculations. The conducted multicriteria analysis is based on AHP [44]. This procedure integrated the standardization of factor values, factor weights determination and the aggregation of selected factors and constraints.

Standardization was performed by stepwise standardization methods in four classes in 1–4 number intervals, where 1 denotes the lowest and 4 denotes the highest impact on the result. This approach allowed the integration of quantitative and qualitative factor types [45], which are common in the multicriteria analysis of crops [46]. Weight determination of the individual factors was performed using pairwise comparison matrices, as part of the AHP method [47]. Pairwise comparison incorporates a scale of preference between every combination of two factors, with preference factors ranging from 1 (equal preference) to 9 (extreme preference). A detailed procedure of pairwise comparison is described in [48]. Its consistency was calculated by the consistency index (*CI*) and consistency ratio (*CR*), as shown in Equations (1) and (2). The random consistency index (*RI*) represents an average *CI* from a random matrix, $\lambda$ is the average value of consistency vector and *n* is the number of parameters [49]:

$$CR = \frac{CI}{RI}, \qquad (1)$$

$$CI = \frac{\lambda - n}{n - 1}.$$

(2)

An acceptable value of inconsistency is $CR \leq 0.10$, while larger values indicate a necessity for modification of pairwise comparisons [50,51]. Aggregation of factors and constraints was conducted using the weighted linear combination. Such a procedure is commonly used in multicriteria analysis for criteria aggregation [52]. A weighted linear combination (*score*) for the calculation of sustainability and vulnerability was calculated using the Equation (3) [53]:

$$score = \sum w_i X_i \times C$$

(3)

where $w_i$ is the weight of the factor $i$, $X_i$ is standardized values of the factor $i$ and $C$ is a Boolean layer of constraints. Sustainability and vulnerability results range from 1 to 4. The ascending nature of these values represents the increase of sustainability and a decrease of vulnerability.

2.2.4. Suitability Calculation Method

The hybridization of the multicriteria analysis procedure was performed by unsupervised classification for suitability calculation. The selected classification algorithm was K-means, which was, in multiple studies, applied in multicriteria analysis [54,55] and various branches of agriculture [56–58]. Classification inputs were sustainability and vulnerability rasters, with equal impact on classification result. Unconstrained pixels were classified into four suitability classes: highest suitability, moderate-high suitability, moderate-low suitability and lowest suitability. Such classification meets the standards set by FAO, in which the land suitability evaluation is generally represented by suitable and unsuitable regions and mainly categorized in the five suitability classes [23]. In this research, four suitable classes were calculated during the unsupervised classification and one permanently unsuitable class is represented by constraints, completing the five class representation. Ranking of the suitable classes was performed based on the suitability value, calculated as an average of mean sustainability and vulnerability class values. These two values are considered as equally impactful on the suitability result, which is the reason that the regular average was selected for the integration of sustainability and vulnerability values.

## 3. Results

### 3.1. Evaluation of Tested Interpolation Methods

The study area was expanded with a 1 km buffer area for soil sampling to avoid extrapolation on sparsely sampled areas. All of the collected 171 samples in the field were analysed in the laboratory for humus, $K_2O$, $P_2O_5$ and pH. Descriptive statistics for all five random training sets are calculated as shown in Table 2, using the Python v3.7.1 (Wilmington, Delaware, United States of America) SciPy library. The mean values of five datasets for all soil features showed moderate variability. The difference of highest and lowest mean value is largest for $K_2O$, and amounts 0.82 mg/100 g. CV values remained consistent for all sets, which, in combination with mean values, indicate moderate data stationarity. High variability of skewness and kurtosis was observed. The values of pH are considered fairly symmetrical due to the mean skewness of −0.43. Humus and $K_2O$ resulted in moderate skewness, with mean skewness values of 0.59 and 0.64, respectively. $P_2O_5$ was determined as highly skewed, having a mean skewness of 1.31. Mean kurtosis values range from −1.29 for pH to 0.90 for $P_2O_5$, implying platykurtic distribution. All mean kurtosis values deviate from normal distribution kurtosis with a value of 3. Considering skewness and kurtosis values of analysed soil features, it was determined that none of them possess normal distribution.

**Table 2.** Descriptive statistics of interpolation methods evaluation training datasets.

| Soil Feature | Value | Set 1 | Set 2 | Set 3 | Set 4 | Set 5 |
|---|---|---|---|---|---|---|
| Humus | Mean | 2.77 | 2.72 | 2.67 | 2.71 | 2.68 |
| | CV | 0.38 | 0.37 | 0.38 | 0.39 | 0.38 |
| | SK | −0.32 | 0.71 | 0.88 | 0.82 | 0.87 |
| | KT | 0.83 | −0.64 | −0.18 | −0.37 | −0.44 |
| $K_2O$ | Mean | 16.81 | 16.56 | 17.38 | 16.57 | 16.96 |
| | CV | 0.48 | 0.44 | 0.43 | 0.43 | 0.44 |
| | SK | 0.85 | 0.65 | 0.61 | 0.56 | 0.55 |
| | KT | 0.95 | −0.08 | −0.19 | −0.25 | −0.39 |
| $P_2O_5$ | Mean | 16.91 | 16.43 | 16.40 | 16.32 | 16.49 |
| | CV | 0.66 | 0.67 | 0.70 | 0.68 | 0.69 |
| | SK | 1.29 | 1.31 | 1.28 | 1.39 | 1.30 |
| | KT | 0.88 | 0.99 | 0.71 | 1.12 | 0.78 |
| pH | Mean | 7.19 | 7.14 | 7.03 | 7.11 | 7.03 |
| | CV | 0.15 | 0.15 | 0.15 | 0.15 | 0.15 |
| | SK | −0.59 | −0.47 | −0.39 | −0.39 | −0.32 |
| | KT | −1.16 | −1.30 | −1.34 | −1.30 | −1.33 |

CV: Coefficient of variation, SK: Skewness, KT: Kurtosis.

Since datasets do not possess a normal distribution, logarithmic transformation was performed as a preliminary step to OK. The variogram model and fitting range were selected based on the internal goodness-of-fit values. Combinations with the highest coefficients of determination between the variogram model and empirical variogram were selected. Tested mathematical models were linear, logarithmic, Gaussian and spherical. The Gaussian model was selected as optimal for the interpolation of humus, $K_2O$ and pH. The spherical model produced the highest goodness-of-fit values for low stationarity $P_2O_5$ samples. IDW was conducted with inverse distance to a power function, with a weighting coefficient of 3. The Sibson method was used for interpolation using NN. Table 3 contains $R^2_{inner}$ and $R^2_{outer}$ values for each tested interpolation combination.

**Table 3.** Interpolation methods accuracy values by datasets and soil features.

| Soil Feature | Interp. | Set 1 | | Set 2 | | Set 3 | | Set 4 | | Set 5 | |
|---|---|---|---|---|---|---|---|---|---|---|---|
| | | $R^2_{inner}$ | $R^2_{outer}$ | $R^2_{inner}$ | $R^2_{outer}$ | $R^2_{inner}$ | $R^2_{outer}$ | $R^2_{inner}$ | $R^2_{outer}$ | $R^2_{inner}$ | $R^2_{outer}$ |
| Humus | OK | 0.821 | 0.785 | 0.876 | 0.680 | 0.794 | 0.853 | 0.826 | 0.772 | 0.878 | 0.619 |
| | IDW | 0.997 | 0.700 | 0.999 | 0.626 | 0.996 | 0.754 | 0.996 | 0.591 | 0.997 | 0.644 |
| | NN | 0.983 | 0.719 | 0.991 | 0.654 | 0.968 | 0.816 | 0.983 | 0.525 | 0.986 | 0.615 |
| $K_2O$ | OK | 0.729 | 0.646 | 0.717 | 0.606 | 0.718 | 0.515 | 0.788 | 0.446 | 0.731 | 0.556 |
| | IDW | 0.993 | 0.803 | 0.992 | 0.792 | 0.992 | 0.684 | 0.991 | 0.503 | 0.992 | 0.821 |
| | NN | 0.992 | 0.795 | 0.991 | 0.600 | 0.984 | 0.707 | 0.985 | 0.472 | 0.985 | 0.758 |
| $P_2O_5$ | OK | 0.852 | 0.315 | 0.682 | 0.374 | 0.732 | 0.153 | 0.700 | 0.255 | 0.795 | 0.292 |
| | IDW | 0.989 | 0.301 | 0.988 | 0.524 | 0.989 | 0.349 | 0.988 | 0.435 | 0.989 | 0.498 |
| | NN | 0.986 | 0.428 | 0.987 | 0.496 | 0.979 | 0.398 | 0.976 | 0.363 | 0.979 | 0.359 |
| pH | OK | 0.850 | 0.650 | 0.800 | 0.725 | 0.896 | 0.552 | 0.794 | 0.748 | 0.789 | 0.767 |
| | IDW | 0.998 | 0.630 | 0.999 | 0.663 | 0.999 | 0.564 | 0.998 | 0.712 | 0.998 | 0.809 |
| | NN | 0.989 | 0.633 | 0.992 | 0.661 | 0.992 | 0.610 | 0.990 | 0.703 | 0.987 | 0.825 |

OK: Ordinary kriging, IDW: Inverse distance weighted, NN: Natural neighbour, $R^2_{inner}$: Inner interpolation accuracy, $R^2_{outer}$: Outer interpolation accuracy.

IDW constantly produced the highest average $R^2_{inner}$ for all soil features, closely followed by NN (Table 4). OK had an average of 0.206 lower $R^2_{inner}$ value than IDW, being the least favourable option based on inner accuracy test. IDW performed best with both $R^2_{inner}$ and $R^2_{outer}$ on samples with lowest stationarity ($K_2O$) data and highly skewed samples ($P_2O_5$). OK resulted with higher $R^2_{outer}$ the more samples were closer to having a normal distribution and high stationarity. An unevenness of these

properties caused a major variability of average $R^2_{outer}$ values for OK, ranging from 0.278 for $P_2O_5$ to 0.742 for humus. NN remained consistent with the second-highest average accuracy values for all tested soil features. Considering the interpolation accuracies of all four soil features, IDW was selected as an optimal interpolation method.

**Table 4.** Interpolation methods accuracy values by datasets and soil features.

| Interp. | Humus | | K$_2$O | | P$_2$O$_5$ | | pH | |
|---|---|---|---|---|---|---|---|---|
| | $R^2_{inner}$ | $R^2_{outer}$ | $R^2_{inner}$ | $R^2_{outer}$ | $R^2_{inner}$ | $R^2_{outer}$ | $R^2_{inner}$ | $R^2_{outer}$ |
| OK | 0.839 | 0.742 | 0.737 | 0.554 | 0.752 | 0.278 | 0.826 | 0.688 |
| IDW | 0.997 | 0.663 | 0.992 | 0.721 | 0.989 | 0.422 | 0.998 | 0.675 |
| NN | 0.982 | 0.666 | 0.987 | 0.667 | 0.981 | 0.409 | 0.990 | 0.686 |

OK: Ordinary kriging, IDW: Inverse distance weighted, NN: Natural neighbour, $R^2_{inner}$: Inner interpolation accuracy, $R^2_{outer}$: Outer interpolation accuracy.

### 3.2. Sustainability and Vulnerability Calculation

Standardization of sustainability factors was conducted as shown in Table 5. Qualitative factors were standardized by assigning a standardized value to each subclass. Standardization of quantitative factors was performed by the reclassification of the selected value interval to new standardized values. Qualitative vulnerability factors were standardized by the binary reclassification of vulnerable classes to 1 and non-vulnerable classes to 4.

Pairwise comparison matrices were created by vegetable cultivation expert for sustainability (Table 6) and vulnerability factors (Table 7). Consistency values show that both pairwise comparison matrices resulted in acceptable consistency. Fertility and dainage factors were considered most influential in a first pairwise comparison, with combined 56.9% of the total impact on sustainability. Private forests with high wood potential resulted as the most vulnerable factor, comprising almost 45% of the total vulnerability impact.

**Table 5.** Standardization of sustainability factor values.

| Factor | Standardized Value | | | |
|---|---|---|---|---|
| | 1 | 2 | 3 | 4 |
| Fertility | Other soils | Eugley soils | Hydroameliorated soils | Semigley soils |
| Drainage | Very low | Low | Moderate | High |
| Humus | <2.5 | 2.5–3.0 | 3.0–3.5 | 3.5> |
| K$_2$O | <5.0 | 5.0–10.0 | 10.0–15.0 | 15.0> |
| P$_2$O$_5$ | <6.0 | 6.0–10.0 | 10.0–20.0 | 20.0> |
| pH | <5.0, 7.3> | 5.0–5.6 | 5.6–6.4 | 6.5–7.2 |
| Slope | 2.0> | 1.5–2.0 | 1.0–1.5 | <1.0 |

**Table 6.** Sustainability pairwise comparison matrix.

| | Fertility | Drainage | Humus | K$_2$O | P$_2$O$_5$ | pH | Slope | Weights |
|---|---|---|---|---|---|---|---|---|
| Fertility | 1 | 2 | 2 | 4 | 5 | 7 | 8 | 0.329 |
| Drainage | 1/2 | 1 | 2 | 3 | 4 | 6 | 7 | 0.240 |
| Humus | 1/2 | 1/2 | 1 | 3 | 4 | 5 | 6 | 0.190 |
| K$_2$O | 1/4 | 1/3 | 1/3 | 1 | 2 | 4 | 5 | 0.105 |
| P$_2$O$_5$ | 1/5 | 1/4 | 1/4 | 1/2 | 1 | 2 | 4 | 0.067 |
| pH | 1/7 | 1/6 | 1/5 | 1/4 | 1/2 | 1 | 3 | 0.043 |
| Slope | 1/8 | 1/7 | 1/6 | 1/5 | 1/4 | 1/3 | 1 | 0.026 |

*n* = 7, *CI* = 0.071, *RI* = 1.32, *CR* = 0.054.

**Table 7.** Vulnerability pairwise comparison matrix.

|   | A | B | C | D | E | Weights |
|---|---|---|---|---|---|---------|
| A | 1 | 2 | 4 | 5 | 8 | 0.446 |
| B | 1/2 | 1 | 3 | 4 | 6 | 0.288 |
| C | 1/4 | 1/3 | 1 | 2 | 4 | 0.132 |
| D | 1/5 | 1/4 | 1/2 | 1 | 4 | 0.095 |
| E | 1/8 | 1/6 | 1/4 | 1/4 | 1 | 0.039 |

$n = 5$, $CI = 0.058$, $RI = 1.12$, $CR = 0.052$.

Individual Boolean constraint criteria layers were multiplied mutually to form a constraints layer. A total of 85.1% of the study area was constrained during criteria aggregation, which resulted in the area of 47 km$^2$ available for red spicy pepper cultivation. Sustainability values ranged from 1.77 to 3.63, indicating moderate sustainability with no extreme values. Vulnerability resulted in the value range of 1.50–4.00, indicating a high variability of vulnerability on the calculated area. A total of 35.6% of the area resulted as non-vulnerable with the most favourable value (4.00).

*3.3. Suitability Calculation*

K-means classification resulted in four classes, with their mean sustainability and vulnerability shown in Table 8. The classification was performed with 20 maximum iterations. The mean suitability values of each class were ranked and the description to each class was added. The highest suitability class was dominantly located in the northwest of the municipality and covered over one-third of the unconstrained area. The top two classes are considered as suitable for red spicy pepper cultivation based on the relative suitability values between classes, covering the area of 2167.5 ha. The bottom two classes had significant limitations in the form of high vulnerability (moderate-low suitability) or low sustainability (lowest suitability). The final red spicy pepper suitability map was shown in Figure 3, alongside sustainability and vulnerability maps.

**Table 8.** Calculation of suitability values from unsupervised classification results.

| Suitability Class | Area (ha) | Mean Sustainability | Mean Vulnerability | Suitability Value |
|-------------------|-----------|---------------------|--------------------|-------------------|
| Highest suitability | 1667.0 | 3.267 | 3.968 | 3.618 |
| Moderate-high suitability | 500.5 | 2.993 | 3.961 | 3.477 |
| Moderate-low suitability | 1330.5 | 3.197 | 3.118 | 3.158 |
| Lowest suitability | 1178.8 | 2.216 | 3.965 | 3.091 |

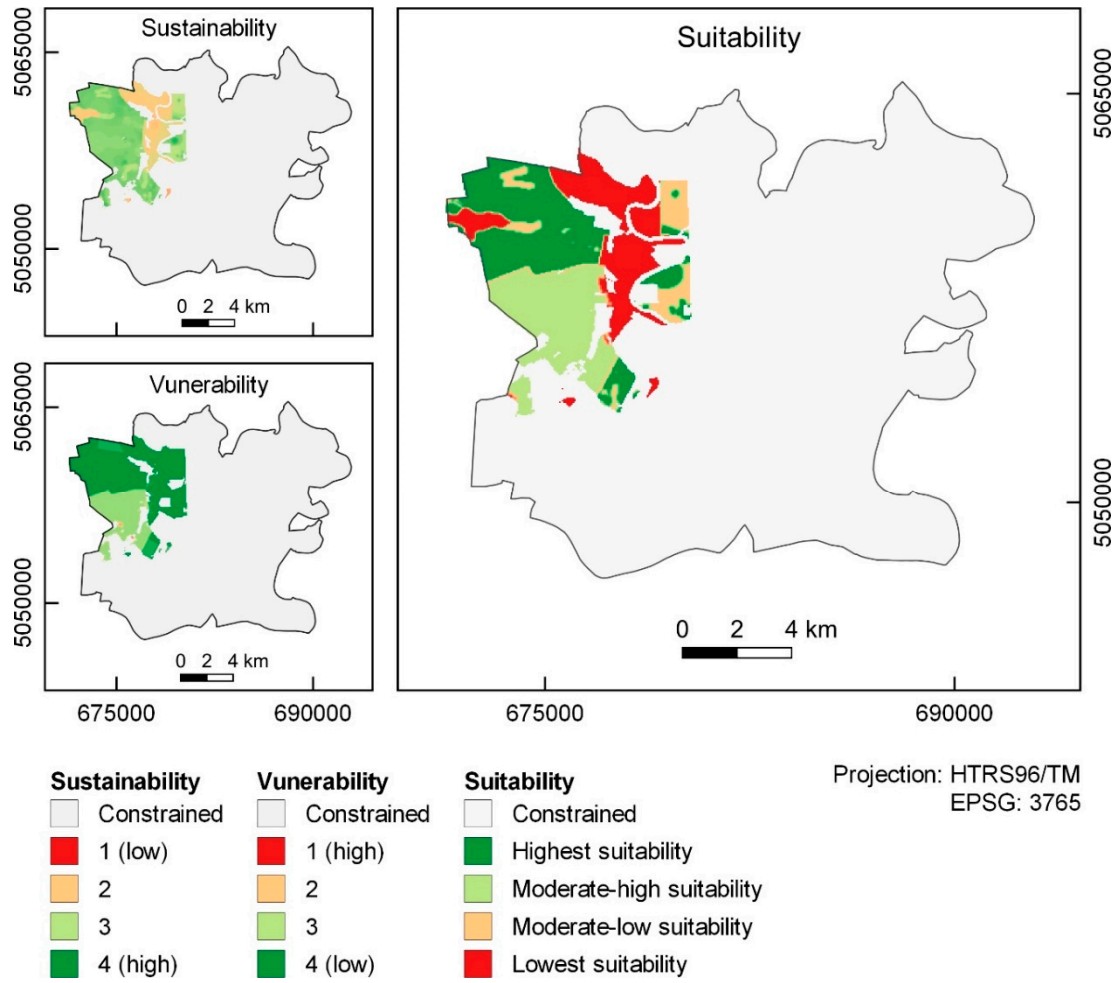

**Figure 3.** Suitability, sustainability and vulnerability maps of the study area for red spicy pepper cultivation.

## 4. Discussion

The versatility of applied methods in the process of suitability calculation ensures the repeatability of the process, not related to the study area size or its location in the world. Existing studies on the application of multicriteria analysis on crop suitability are developed in similar areas, like the study area in this research, and imply the applicability of this methodology on larger areas [15,35,59–61]. The same studies noted the flexibility of the used methods regarding the input data selection. Thus, the addition or deduction of new individual criteria or the criteria group is supported by the applied methods. Consequentially, the applicability of the suitability calculation proposed in this study is possible on almost any crop type, with the prerequisite of the input criteria adjustment to the selected crop type and the specificities of the study area. Evaluation of interpolation methods produced results that are contrary to multiple studies in which kriging methods resulted in better interpolation accuracy than deterministic methods [62,63]. The primary cause of the better accuracy of IDW than OK in this study is the low normality and moderately low data stationarity. Considering that OK produced higher accuracy, the closer the sample data are to a normal distribution, it is assumed that OK would be optimal interpolation method for the majority of samples in practice. Nevertheless, this case demonstrated that the evaluation of the interpolation method was an important step in the multicriteria analysis and that kriging is not always superior to deterministic methods. Evaluation of interpolation methods can be expanded using additional statistical indicators like root mean square error and mean relative error [64,65].

The climate criteria group, consisting of air temperature, precipitation and solar radiation, were not applied to this analysis due to homogenous climate conditions in the study area. However, in the

case of more diverse conditions, these data have a major impact on the suitability results [66]. Similarly, the addition of the soil nitrogen content was strongly considered as a sustainability factor, but was dismissed due to its high homogeneity in the study area. The potential importance of soil nitrogen content as a fundamental macronutrient for red spicy pepper suitability calculation increases with its variability in the study area, presenting a base of future research. Stepwise standardization allowed better control and more expert influence on standardization in contrast to common linear scaling. This is primarily obvious in the case of a few extremely high or low pixel values. Fuzzy membership methods for standardization offer a potential upgrade to standardization [67] and will be a subject of future research. One of the main disadvantages of AHP is an expert's subjectivity in the creation of pairwise comparison matrices, which could be a source of inaccuracies in the suitability result [68]. These inaccuracies can be reduced in the form of fuzzy AHP [69] or by aggregating estimations of multiple experts [14].

The separation of input criteria to sustainability and vulnerability factors enabled the calculation of positive and negative impacts of red spicy pepper production. The number of input criteria for red spicy pepper cultivation can easily grow and cause cluttering since it includes soil, climate, infrastructure and economic criteria groups. Such a procedure reduces the overemphasis of positive or negative influences and prevents redundancy of factors in the individual analysis [70]. Separation of sustainability and vulnerability allows the limiting of the factors to seven per analysis, which is the maximum recommended number of input criteria in AHP [71]. It is also considered that reducing of the pairwise matrices' complexity leads to a decrease in possible inaccuracies during weight determination. During that procedure, the expert's subjectivity is also restricted, which reduces one of AHP's main drawbacks. The application of unsupervised classification for suitability calculation allowed objective and rule-based calculation, further reducing the probability of error due to expert's subjectivity [72].

The difference in suitability values of the highest and lowest-ranked class is relatively small, 0.527 out of a possible 3.000. Classes were ranked relatively with regard to the mutual relationship of class suitability values. Absolute suitability scales like FAO land-crop suitability scale [73] were considered inappropriate since the distinctively designate top and bottom classes as completely suitable or non-suitable. Further research is planned on suitability result validation with ground-truth data and the measurement of the calculated suitability impact on potential economic profits. Agricultural plots with similar cultivation methods are required for planned research, alongside yield data for each observed plot.

## 5. Conclusions

The proposed methodology was based on the hybridization of multicriteria analysis and unsupervised classification for the calculation of red spicy pepper cultivation suitability. One of the main advantages of the proposed methods is the interpolation methods evaluation, as four soil feature criteria required interpolation as part of the pre-processing. It was observed that the IDW method performed best out of the three tested interpolation methods, contrary to most cases where OK had an advantage over deterministic methods. Another advantage of methodology was the separation of sustainability and vulnerability multicriteria analysis. These results were used for the calculation of suitability, but can be used independently if necessary. The criteria separation procedure prevented criteria redundancy by limiting the number of criteria for each analysis to seven. It also reduced inaccuracies in calculation due to expert subjectivity. The final advantage represents an aggregation of sustainability and vulnerability results in suitability classes using unsupervised classification. That procedure further reduced errors caused by an expert's subjectivity. It was also used to create ranked classes that are easily applicable in the planning of new red spicy pepper plantations. With the use of unsupervised classification, 2167.5 ha (6.9% of the total study area) was determined as suitable for red spicy pepper cultivation. Further research is planned on the validation of the calculated results using ground-truth data and the integration of fuzzy methods with the multicriteria analysis procedure.

**Author Contributions:** Conceptualization: M.J. and I.P.; methodology: M.J. and I.P.; validation: M.J. and I.P.; formal analysis: D.R.; investigation: O.A.; resources: O.A.; data curation: D.R.; writing—original draft preparation: D.R.; writing—review and editing: M.J. and I.P.; visualization: D.R. All authors have read and agreed to the published version of the manuscript.

**Funding:** This research received no external funding.

**Acknowledgments:** This work was supported by the Faculty of Agrobiotechnical Sciences Osijek as a part of the scientific project 'AgroGIT–technical and technological crop production systems, GIS and environment protection'.

**Conflicts of Interest:** The authors declare no conflict of interest.

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
