# Peer review of "Suitability Calculation for Red Spicy Pepper Cultivation (Capsicum annum L.) Using Hybrid GIS-Based Multicriteria Analysis"

_agronomy, doi:10.3390/agronomy10010003_

Round 1

Reviewer 1 Report

Thank you for the thorough consideration of my comments, and the excellent additions to the manuscript.

Reviewer 2 Report

Overall, I am happy with the revisions made by the authors. It should be noted that, as I commented on the earlier version, this paper still needs edits and careful check of the text for improved quality of English writing. There are a number of grammatical and wording errors. In particular, please correct the inconsistencies (use low-case numbers) of the chemical formulae in Section 3.1 and Table 6, and define the abbreviations for the interpolation methods in Tables 3-4. The authors may ask a qualified person in English to proofread the paper during next iteration of revisions. 

Author Response

This manuscript is a resubmission of an earlier submission. The following is a list of the peer review reports and author responses from that submission.

Round 1

Reviewer 1 Report

This paper present an evaluation of land suitability based on GUI and multi-criteria analysis. Overall it still needs polishing up in writing to be streamlined, and eliminating grammatical or wording errors, and the experimental section needs be clearly described.  Here are my specific comments:

Since constraints are factored in the sustainability calculation, in Introduction the author may also discuss how this factor along with sustainability and vulnerability was applied in the previous research. Line 43-44, please expand a little further on how unsupervised classification allows reducing man-made errors in suitability calculation. Line “This paper aims to present the methodology of …”. The use of GIS-based multi-criteria analysis for land suitability assessment is not something new. The authors may highlight the major objectives and contributions of the paper, rather than just “present” the existing methods. Line 90. “Defined criteria for … in Table 1”. This sentence is rather confusing. Are you trying to define some ad-hoc criteria for this study or just choose from those defined or used in literature? It is noted that the majority of these criteria are descriptive. How were they quantified or quantitatively described? For instance, soil fertility is an integrative attribute that involves a multitude of physicochemical properties. Lines 115-116 is not clear. Please rephrase it. Regarding interpolation and classification methods, were all the experimented methods implemented with the build-in functionality of the GIS software? Lines 124-125, “in five independent repetitions”, how was those repetitions achieved? It is noted that data was partition with a ratio of 7:3 for training and test (Lines 125-126). But in the later section Lines 178-179, why 120 training and 51 test samples? Line 162 how was the weight determined? Lines 170-171, how were the four classes of suitability decided? Be more specific? Line 172, “calculated as an average of …”, why taking the average? Any rationale behind? Table 2, notations should be given to the abbreviations (to keep tables self-explanatory, the same thing with other tables)

Reviewer 2 Report

The title of the manuscript (MS) deals with Suitability calculation for red spicy pepper cultivation (Capsicum annum L.) using hybrid GIS-based multicriteria analysis. While the topic is potentially important, the manuscript (MS) needs more work to be considered for publication.

I suggest the authors consider my comments in the attached PDF. Thanks.
